# Current Methods and Pipelines for Image-Based Quantitation of Nuclear Shape and Nuclear Envelope Abnormalities

**DOI:** 10.3390/cells11030347

**Published:** 2022-01-20

**Authors:** Anne F. J. Janssen, Sophia Y. Breusegem, Delphine Larrieu

**Affiliations:** Department of Clinical Biochemistry, Addenbrookes Biomedical Campus, Cambridge Institute for Medical Research, University of Cambridge, Cambridge CB2 0XY, UK; afj28@cam.ac.uk (A.F.J.J.); syab2@cam.ac.uk (S.Y.B.)

**Keywords:** nuclear envelope, nuclear envelope abnormality, lamina, nuclear morphology

## Abstract

Any given cell type has an associated “normal” nuclear morphology, which is important to maintain proper cellular functioning and safeguard genomic integrity. Deviations from this can be indicative of diseases such as cancer or premature aging syndrome. To accurately assess nuclear abnormalities, it is important to use quantitative measures of nuclear morphology. Here, we give an overview of several nuclear abnormalities, including micronuclei, nuclear envelope invaginations, blebs and ruptures, and review the current methods used for image-based quantification of these abnormalities. We discuss several parameters that can be used to quantify nuclear shape and compare their outputs using example images. In addition, we present new pipelines for quantitative analysis of nuclear blebs and invaginations. Quantitative analyses of nuclear aberrations and shape will be important in a wide range of applications, from assessments of cancer cell anomalies to studies of nucleus deformability under mechanical or other types of stress.

## 1. Introduction

While multiple cellular factors are required to maintain nuclear genome integrity and organisation, a primordial role is played by the nuclear envelope (NE). The NE (reviewed in [1,2,3]) consists of a double lipid bilayer that envelops the nucleoplasmic material (Figure 1a). It is perforated with “pores”, formed by the nuclear pore complexes (NPC), which ensure the transport of proteins and RNA between the nucleoplasm and cytoplasm. Linker of Nucleoskeleton and Cytoskeleton (LINC) complexes that span the inner (INM) and outer nuclear membrane (ONM) link the NE to the cytoplasmic cytoskeleton, enabling the nucleus to function in mechano-transduction. In metazoans, a network of lamin intermediate filaments lies on the inside of the NE envelope which not only provides mechanical support to the NE, but also organises chromatin and links with multiple nucleoplasmic regulators, thereby playing important roles in the regulation of diverse cellular processes, including DNA replication and repair and gene expression control. Chromatin itself (the DNA and its associated proteins) also contributes to the mechanical stability of the nucleus [4]. Embedded in the INM are proteins from the LEM (LAP2-emerin-MAN1) domain family that all bind to Barrier-to-Autointegration Factor (BAF), a DNA and lamin binding protein which plays import roles in nuclear envelope integrity.

The different components of the NE act together to guard the integrity of the nucleus, i.e., to protect the genome and respond to changing forces acting upon it. Indeed, the nucleus is not a rigid compartment, and the nuclear shape can vary drastically depending on forces acting on the NE, either from the outside (e.g., cytoskeleton) or from within, as well as on tension that can occur within the NE [5]. For example, cells moving through constricted spaces (such as metastatic cancer cells travelling through tiny blood vessels) undergo dramatic nuclear shape changes [6,7,8,9]. These changes also accompany gene regulatory events such as chromatin decondensation during transcription regulation. The most dramatic nuclear shape changes occur when cells undergo open or closed mitosis.

Beyond global shape changes, the nucleus can also present other types of NE anomalies such as NE invaginations, blebs and ruptures (Figure 1b). Importantly, while all of these changes to the nuclear integrity occur at a low frequency in normal, healthy cells, they can dramatically increase in conditions where the NE integrity becomes challenged. This can occur upon mechanical stress or when the NE structure is weakened by depletion or mutations of NE proteins. The latter occurs in several laminopathies such as Emery Dreifuss Muscular Dystrophy (EDMD) [10,11,12,13,14] and in progerias (premature aging disorders) including Hutchinson-Gilford Progeria Syndrome (HGPS) [15,16,17]. Nuclear irregularities also occur in many cancer types, and visual inspection of the nuclear shape in cancer biopsies is widely used to assess both cancer stage and prognosis [18,19,20].

For these reasons, the detection and quantification of nuclear shape and NE alterations can have broad applications, from cell biologists investigating mechano-transduction or cell migration through constricted spaces, to pathologists determining disease diagnosis or severity. This review focusses on current methods and pipelines to quantify NE defects in interphase cells—although some of the described anomalies might originate from imperfect NE assembly after mitosis. The methods described here mostly rely on the analysis of fluorescence microscopy images, but several of these can be adapted for use with histology stains. We present methods and pipelines to quantify nuclear shape deviations from the canonical round nuclear shape (circular shape in 2D and spherical shape in 3D) and NE anomalies such as micronuclei, NE blebs and invaginations. The focus will mainly be on 2D analyses, as these are most commonly used and are most accessible. We illustrate these methods with examples and present new analysis pipelines for use both with “regular” fluorescence microscopy images and in high-content screening applications.

## 2. Quantitative Analysis of Nuclear Shape

### 2.1. Nuclear Labelling

In order to analyse nuclear shape quantitatively, it is paramount to achieve good nuclear labelling. This should allow not only for a good definition of the nuclear contour, but also for the accurate identification of the individual nuclei as objects to analyse (segmentation). Focussing on fluorescent labelling, many researchers are familiar with organic DNA-binding fluorophores (e.g., DAPI, Hoechst 33342, Hoechst 33258), which become highly fluorescent upon binding to the DNA minor groove in a sequence-independent manner [21,22]. However, in many instances, it is recommended to include a staining of the NE (using antibodies against proteins such as nuclear pore complexes, LINC complex etc.) or nuclear lamina (Lamin A/C or Lamin B). This not only much better defines the nuclear/cytoplasm boundary, but also allows the analysis of other nuclear abnormalities beyond shape such as invaginations, blebs etc. [23] (see below). In most cases, a UV-absorbing DNA stain can be combined with antibody-labelling of the nuclear lamina or a NE-resident protein to allow for both nuclei identification, segmentation and selection (all based on the DNA stain) as well as nuclear shape analysis (using either the NE/lamina stain or both stains). Inclusion of a DNA stain has the specific advantage of allowing for the exclusion of the nuclei in mitosis, based on their increased DNA staining intensity. Of note, many segmentation algorithms give an option to smooth (i.e., locally average) measured fluorescence intensities. For each analysis this should be carefully considered, as smoothing generally removes irregularities in the nuclear contour and could therefore remove them from the nuclei as they are selected for analysis.

### 2.2. Quantitative Parameters to Analyse Nuclear Shape

Early reports of irregular nuclear shape generally relied on manual visual classification, a method inherently subject to observer bias and therefore liable to poor reproducibility. This was sometimes mitigated by including several observers blinded to the experimental details. Nevertheless, a quantitative and automated analysis of nuclear shape was clearly needed to avoid this bias and increase the throughput.

While not strictly describing nuclear shape, it is worth noting that a number of quantitative parameters of nuclear size can also indirectly reflect the presence of NE defects and are often easier to compute in commonly used image analysis software packages. These include nuclear area, nuclear diameter and nuclear perimeter. For example, mouse embryonic fibroblasts derived from mice with a H222P mutation in lamin A, which in humans causes dilated cardiomyopathy, were shown to have larger nuclei compared to their wild-type counterparts [24]. Alternatively, knockdown of ELYS, a protein of the nuclear pore complex, causes smaller nuclei which was proposed to be correlated to decreased nuclear transport capacity [25]. However, nuclear size changes can also be influenced by many other factors such as chromatin compaction state, cytoskeletal organisation, cell cycle phase etc. It is also worth noting that nuclear size in 2D might not be a good indicator of total nuclear volume. For example, a flattened nucleus would display an increased nuclear area in 2D compared to the same nucleus in a rounded state. This can be influenced by extracellular factors such as substrate rigidity, cell confluency, time since plating etc.

#### 2.2.1. Nuclear Roundness

The most commonly used quantitative measure of nuclear shape is the ratio of 4π times the nuclear area (A) to the nuclear perimeter (P) squared (4πA/P^2^) (Figure 2a). This is a measure of the roundness of the nucleus; for a perfectly circular nucleus, the ratio is 1, while for a more deconvoluted shape (e.g., with blebs or lobulations), the ratio is smaller than 1. This ratio has been named the nuclear contour ratio [11], nuclear circularity [26] or form factor, e.g., in the open source image analysis package CellProfiler [27]. In some image analysis software packages, the inverse of this ratio is used for analysis of an object’s roundness; in this case the analysed shape’s ratio is 1 for a perfect circle and increases when the nucleus deviates from a circular shape. Nuclear circularity analysis has been used to analyse the effects of NE dysfunction in various cellular contexts such as cancer [26,28], laminopathies such as EDMD [11] and HGPS [17,29] or cardiomyopathies [30]. Another easy to compute ratio indicative of nuclear roundness is the nuclear length-to-width ratio or axis ratio. More frequently a related nuclear eccentricity parameter is calculated instead [31,32]. Eccentricity is a measure of the shape of the bounding ellipse; a circle has an eccentricity of 0, and a more elongated shape is associated with a higher eccentricity (Figure 2a).

#### 2.2.2. Boundary Curvature and Solidity

After defining the nuclear contour, one can also characterise boundary curvature. This is a very good method to quantify nuclear shape as curvature provides a complete description of nuclear shape. Indeed, defining the mean negative curvature as the average of all the concave curvatures on the boundary of a nucleus, could accurately distinguish the highly-blebbed nuclei in fibroblasts from HGPS patients from the similarly-sized nuclei in healthy donor fibroblasts [31]. In addition, the same paper showed that the mean negative curvature was not dependent on imaging intensity or cell density. Moreover, the highly lobulated or deconvoluted nuclear shape of HGPS cells could also be accurately characterised through its solidity, the ratio of the nucleus’ measured area to the area of its convex hull shape (Figure 2a).

#### 2.2.3. Elliptic Fourier Transform

An alternative measure of nuclear shape uses an elliptic Fourier transform, approximating the nuclear shape as a sum of harmonic ellipses. This analysis has been used in 2 ways, either determining the number of ellipses required to accurately describe the nuclear shape [11], or using a fixed number of ellipses and making a ratio of their coefficients. The latter was used in a recent RNAi screen for epigenetic modulators of nuclear shape in breast cancer epithelial cells [33]. The authors also computed the solidity of their measured nuclei and demonstrated that the elliptic Fourier coefficient ratio measurements showed a larger dynamic range and are consequently more sensitive to nuclear shape deformations.

#### 2.2.4. Nuclear Symmetry

Looking at nuclear symmetry can also be informative. Nikonenko and Bozhok (2015) compared two measures of nuclear symmetry: radial asymmetry and point asymmetry. Radial asymmetry determines the number of pixels outside the largest inscribed circle of a nucleus. Point asymmetry quantifies the number of pixels that do not have a symmetric pixel partner with respect to the nucleus’ centroid. They found that compared to the circularity measurement, the symmetry-based analyses were better at distinguishing oval shapes versus other irregular shapes [34]. The authors have made their analysis available via an ImageJ plugin.

#### 2.2.5. Comparison of Shape Descriptors

In summary, there are many quantitative nuclear shape parameters that have different sensitivities towards specific types of shape aberrations; as such, a single quantitative shape analysis might not necessarily reflect an observed nuclear shape change (Figure 2). Sometimes, more than one analysis is required to convey the observed nuclear shape abnormalities. In addition, results can depend on how well the nuclear contour is defined; for example, a set of densely packed NE bulges or blebs might not change the overall nuclear roundness significantly, and might thus not be reflected in quantitative parameters of nuclear roundness. Figure 2 illustrates this with analysis of DAPI-stained nuclei in fibroblasts derived from a patient with Nestor-Guillermo progeria syndrome (NGPS) (Figure 2b,c) and lamin-stained nuclei of control and HGPS fibroblast cells (Figure 2d,e). The control and HGPS nuclei examples in Figure 2d have very similar values for the three most commonly used nuclear shape measures (solidity: 0.99 vs. 0.97, form factor: 0.86 vs. 0.81 and eccentricity: 0.73 vs. 0.67). However, analysis of a larger number of cells does detect a significant difference for the nuclear solidity and eccentricity (Figure 2e).

While we have so far focused on nuclear shape analysis in fixed cells, quantitative analysis of nuclear shape dynamics is possible in time-lapse imaging experiments in live cells. Usually, a genetically-encoded fluorescent nuclear reporter (e.g., a GFP-labelled histone protein) is used to follow the nuclear contour over time. Some DNA stains are also compatible with live-cell imaging [35] and fluorescent protein labelling of lamins or NE transmembrane proteins can also be used. Looking at the flickering of the nucleus, i.e., the changes in the nucleus’ contour on the time scale of seconds, Chu et al. [36], were able to quantify the mean square amplitude of the fluctuations and showed that this measure correlates with cell cycle stage. Quantitative analysis of nuclear flickering can also yield NE properties such as the NE bending modulus [37].

### 2.3. Note on 3D Analysis

As access to confocal fluorescent microscopes has become more widespread, including in high-throughput settings, there has been a growing interest in defining nuclear morphology in 3D. The open-source image analysis software ImageJ contains a basic 3D volume analysis module (“3D object counter”), while a recently updated ImageJ plugin, NucleusJ, can calculate 15 nuclear parameters from a z-stack of nuclei images, including 3D shape descriptors such as flatness, elongation and sphericity [38,39]. While we believe that these tools are sufficient for most applications, current research aims at refining the 3D shape representation derived from 3D voxels by incorporating 3D surface modelling in the analysis [40].

## 3. Micronuclei

Micronuclei (recently reviewed in [41,42,43]) are small, extranuclear chromatin bodies surrounded by a NE. They arise from defects in chromosome segregation at the end of mitosis, with a lagging chromosome or chromosome fragment being encapsulated in its own NE membrane. Micronuclei are prone to rupture which can lead to genome instability through dramatic DNA rearrangement events (e.g., chromothripsis or chromosome shattering) [44,45]. In addition, exposure of micronuclear DNA to the cytoplasm can activate the cyclic GMP-AMP Synthase (cGAS), which in turn activates the cGAS-stimulator of interferon genes (STING) pathway and leads to expression of proinflammatory genes [46,47]. The presence of micronuclei has been associated with various pathologies such as cancer [42], and the quantitation of the fraction of nuclei with a micronucleus can give an estimate of genome stability.

Micronuclei appear as circular spots in fluorescent microscopy micrographs stained either for DNA, lamin proteins or NE transmembrane proteins such as the lamin B receptor or emerin. We have also found that the nucleoplasmic LEM-domain protein LAP2α is present in micronuclei (Figure 3). However, several nuclear pore complex components were reported to be largely absent from micronuclei [48] and as such the micronuclear NE is not identical to the envelope of the primary nucleus.

Quantification of the fraction of cells containing micronuclei requires analysis of a large number of cells as the occurrence of micronuclei is low in healthy cells (0.5–2.5 micronuclei per 1000 cells) [49]. However, determining micronuclei presence in fluorescent micrographs is relatively straightforward, and they are clearly visible when stained for the DNA within them (e.g., using DAPI or Hoechst 33258) or a NE, nuclear lamina or nucleoplasm component such as emerin, LMNA or LAP2α. Using LAP2-stained images acquired on a high-content high-throughput microscope (Thermo Scientific CellInsight) and its associated software (HCS Studio), we can indeed detect micronuclei (Figure 3). The analysis algorithm first identifies nuclei and potential micronuclei using intensity and morphological features. While some of these are standard (e.g., smoothing, thresholding and segmentation), others are specifically included to aid in the identification of micronuclei. For example, there is a nucleus “cleanup” function that sequentially erodes and dilates the nuclear mask to clear the nucleus of small objects. Cytoplasmic boundaries of the cells are either defined using a whole cell stain (imaged in an additional channel) or approximated through dilation of the nuclear masks. Finally, nuclear objects are classified as a nucleus or micronucleus based on their comparative size within an intact cell, with the maximum micronuclear diameter typically set at a third of the nucleus’ diameter. Measurement of micronuclear frequency, i.e., the percentage of nuclei that have one or more micronuclei associated with them, in an hTERT immortalised fibroblast cell line indicates that 12% of cells have a micronucleus (Figure 3b). Other outputs of the analysis not shown include average micronuclear size as well as micronuclear intensity and morphology measurements.

## 4. Nuclear Envelope Invaginations

NE invaginations, also referred to as the nucleoplasmic reticulum (NR), can be found in many different cell types. Invaginations of the NE are present in healthy cells and tissues [50] and they are thought to play a role in genome organisation, lipid metabolism, DNA repair [51] and regulation of nuclear Ca^2+^ signalling [52,53,54]. Invaginations can range from a few hundred nm’s long to transecting the entire nucleus and are classified into two types. Type I invaginations only involve the invagination of the inner nuclear membrane (INM) while type II invaginations consist of both inner and outer nuclear membrane [55]. Type II invaginations thus have a cytoplasmic core that can contain mitochondria, vesicles, endoplasmic reticulum (ER) like membranes or cytoskeletal components [56,57]. These two types of invaginations can coexist within the same nucleus and type I invaginations can branch off type II invaginations forming complex branched networks. A lamina coat has been detected on both types of invaginations and therefore the presence of intranuclear lamina is often used to detect the presence of invaginations using light microscopy techniques.

While the exact mechanisms underlying invagination formation and stabilisation still need further exploration, formation can occur in interphase and does not merely represent a remnant of faulty post-mitotic NE reassembly. Furthermore, formation of NE invaginations is dependent on synthesis of new proteins and membrane phospholipids and thus doesn’t reflect a reorganisation of the existing nuclear envelope [55,58,59].

Although NE invaginations are observed in healthy cells, their frequency has been observed to increase in various pathologies such as cancer [60,61], tauopathies [57,62], Hutchinson-Gilford Progeria Syndrome (HGPS) [63] and Emery Dreifuss Muscular Dystrophy (EDMD) [64]. Therefore, the development of sensitive and automated tools to quantify these invaginations can have broad applications for both fundamental research and clinical diagnosis purposes.

### 4.1. Quantification of NE Invaginations

The presence of NE invaginations can be visualised using Electron Microscopy (EM) [55,65,66,67] or light microscopy techniques [58,62,67]. Using light microscopy, invaginations can be detected by the presence of lamina, which underlies the inner nuclear membrane, or the presence of a nuclear envelope marker within the nuclear interior. Visual estimation and manual quantification of the number of invaginations is often employed [55,58,66,67], but this is time consuming and can be biased.

Automated quantification of the presence of NE markers in the nuclear interior can therefore be a powerful tool, to give qualitative information on the presence of invaginations, but also quantitative data on the number of these invaginations. A custom FIJI plugin was previously developed [57] which relies on the quantification of lamin B in the nuclear interior using confocal microscopy images. The authors used this quantification to show an increased number of nuclear invaginations in human Fronto-Temporal Dementia (FTD) neurons caused by abnormal microtubule organisation due to mislocalised Tau [57].

Using an analysis pipeline which can detect areas with high levels of lamin B in proximity to the NE or in the nuclear interior, one can estimate the number of invaginations. This can give information on the severity of the invagination phenotype or help to set a threshold to define an invagination positive population. Using an image analysis pipeline we have developed in Cell Profiler [27,68] (Figure 4a), we were able to detect and quantify the amount of intranuclear lamin B in confocal images. In short, the pipeline detects the nucleus based on DAPI staining after which it detects areas with high levels of lamin B. The nucleus object is shrunken by a defined number of pixels allowing the separation of lamin B present at the NE and lamin B localised in the nuclear interior (invaginations) (Figure 4a). Using this pipeline, we could show an increased number of invaginations in a Nestor Guillermo Progeria Syndrome (NGPS) patient cell line, in which NE integrity is compromised (Figure 4b,c). Although this does not seem to be a consistent phenotype of NGPS (Figure 4b,c), we believe that this Cell Profiler pipeline (or similar analysis pipelines) could be of great use for the automated detection of NE invaginations in other cellular and/or disease contexts.

### 4.2. D Imaging of NE Invaginations

Most analyses of NE invaginations are based on 2D information, and therefore, might not only quantify invaginations, but also large folds in the nuclear envelope. In the context of a diseased state, the distinction might not appear that important, although NE folds could represent different implications and molecular causes. To better map the exact organisation of NE invaginations, 3D imaging can be used [54,56,69]. However, analysis of 3D data mostly relies on manual quantification. 3D based analysis of nuclear area occupied by NE invaginations has been done previously, although the exact analysis pipeline remains unclear [54]. Also, a direct comparison of a manual estimation of the number of invaginations using a single confocal slice or a z-stack showed that although absolute numbers might differ, the fold change was similar [66]. Therefore, 2D imaging will provide enough information and will be more efficient when analysing large data sets. It is also more amenable to the analysis of clinical samples as 3D imaging on patient biopsies is challenging [70] and not routinely performed.

## 5. Nuclear Envelope Blebs and Ruptures

Until recently, it was assumed that loss of NE integrity occurred only during mitosis. However, recent work has shown that loss of nuclear compartmentalisation can also happen in interphase nuclei. The loss of NE integrity has indeed been observed in vitro in cells from laminopathy patients [71], in cancer cells [72] and in cells in which the NE had been weakened by viral infection or viral proteins expression [73,74]. In vivo, NE ruptures have been observed as cells experience confined migration through dense tissues [6,7,75], at the invasive edge of human tumours [76] and in aortic smooth muscle cells of a mouse progeria model [77]. In this context, NE ruptures can occur when pressure on the nuclear envelope becomes so high that this leads to a transient loss of NE integrity.

Typically, nuclear envelope rupture is preceded by the formation of a weak point in the nuclear lamina as the nuclear lamina plays an important role in handling nuclear mechanical stress. The lamina gaps are devoid of lamin B1 [6] and often lack other NE proteins such as nuclear pore complexes and LINC proteins [78,79]. Both the presence and absence of Lamin A have been reported at these gaps which might reflect different stages of the rupture and repair process [80,81].

A gap in the lamina allows the formation of a protrusion of the nuclear membrane that temporarily releases the intranuclear pressure. However, under continued stress, the membrane bleb will eventually rupture as the absence of the lamina makes it vulnerable [82]. Nuclear membrane rupture leads to the exposure of chromosomal DNA to the cytoplasm and exchange of material between the cytoplasmic and nucleoplasmic compartments. This can lead to genome instability but also to the activation of innate immune signalling pathways. Maybe not surprisingly, cells can efficiently repair the NE after rupture. Resealing of the NE typically occurs within minutes but can occasionally last for hours. The repair process is mediated by multiple proteins including barrier-to-autointegration factor (BAF), LEM domain proteins and ESCRT-III components [6,7,83,84,85] and is thought to involve the recruitment of ER sheets to the sites of rupture to reseal the membrane.

### 5.1. Analysis of Nuclear Envelope Blebs

The presence of nuclear blebs is thus indicative of a defect in the structural integrity of the NE that can make them prone to ruptures. Therefore, the analysis of nuclear bleb frequency can serve as a means to detect a lack of NE integrity. Typically, blebs are identified based on visual inspection of nuclear morphology by DAPI staining. So far, analysis of bleb frequency mostly relied on manual assignment of nuclei showing blebs.

An alternative method to distinguish nuclear blebs relies on the fact that blebs typically lack lamin B. This lack of lamin B can be used for automated detection of nuclear blebs using an image analysis pipeline. This involves the detection of a chromatin object based on DAPI staining and a lamin B object. Substraction of both areas leads to the definition of a chromatin area lacking lamin B staining (Figure 5a). Care must be taken in setting the intensity threshold of the DAPI image as the blebs often contain less DNA than the rest of the nucleus. Moreover, this analysis can only detect chromatin-containing blebs while blebs lacking chromatin have also been reported [7,80].

Based on this principle, we designed a CellProfiler pipeline (Figure 5a) that can automatically detect and quantify blebs and is therefore useful for the analysis of large image data sets. This pipeline can be used to detect the percentage of cells with blebs, the average size of blebs or could be used to analyse and quantify the recruitment of specific proteins of interest to the bleb sites (Figure 5b,c). Analysis of protein recruitment to blebs can be combined with rupture reporters (see details below) to ensure the blebs analysed have already undergone rupture.

Alternatively, we have also been able to define nuclear blebs in images acquired on a high-content high-throughput microscope using the same algorithm that we used to detect micronuclei but with different constraints (Figure 5d). This analysis is thus not based on the absence of lamin B staining but on morphological features. In particular, the nucleus cleanup function is set to its maximum number of iterations such that blebs are “cut off” from the main nuclear shape and can be analysed; at the same time the “objects” (blebs) need to be touching the main nucleus. Additional restraints can be set, such as object size or circularity to filter out micronuclei that are touching the nucleus as micronuclei are typically smaller and more circular than nuclear blebs. Figure 5d illustrates the detection output of the algorithm, with nuclear blebs indicated by the yellow overlay while micronuclei, indicated in orange, are excluded from the analysis. Quantification of nuclear bleb frequency in hTERT immortalised fibroblasts shows that 10% of nuclei have a nuclear bleb (Figure 5e).

### 5.2. Nuclear Envelope Rupture Analysis

To analyse the loss of nuclear compartmentalisation occurring upon NE rupture, fluorescent proteins fused to a nuclear localisation signal (NLS) or nuclear export signal (NES) are frequently used. Alternatively, tracking the nuclear influx of proteins or molecules that are normally excluded from the nucleus has been used as a marker of rupture e.g., Hsp90 [10], alpha-Tubulin [86] or 70 kDa dextran [71].

The most common reporter of NE rupture is based on (multiple) fluorescent proteins with an NLS signal (e.g., NLS-GFP) [71,72,87]. This reporter accumulates in the nucleus and upon NE rupture can transiently leak into the cytoplasm. Upon repair of the NE, re-accumulation of the fluorescent signal in the nucleus is observed. Thus, this reporter also provides information about repair kinetics in addition to rupture frequency. NE ruptures can also be identified based on accumulation of proteins such as BAF or cGAS at sites where chromatin has been exposed to the cytoplasmic environment [6,7,83]. Overexpression of fluorescently tagged catalytically inactive cGAS, which accumulates on chromatin at the rupture location, can also be used to detect ruptures [7,10] and is usually analysed manually.

Analyses of NLS-GFP localisation in live cell imaging can be done manually, but automated tracking has increased the throughput of this approach [84]. In this manuscript, Robijns et al. used an analysis pipeline written for FIJI image analysis software. The script is based on nuclei detection after image pre-processing followed by object tracking based on a nearest neighbour algorithm and finally track analysis. Usually, co-expression of a nuclear marker that does not leak upon NE rupture, such as mCherry-H2B or a chromatin dye, can be used for object tracking while simultaneously analysing fluctuations in GFP-NLS intensity to detect the presence of ruptures. The authors used this automated analysis to show that rupture frequency is inversely correlated with lamin A/C levels.

A similar automated analysis pipeline was developed in MATLAB to analyse cells migrating through restrictions using microfluidic devices [88]. This algorithm uses several processing steps to identify single nuclei based on fluorescently tagged H2B. Restrictions are localised based on transmitted light images of the device and detection of the circular pillars. Nuclei are subsequently tracked over time including when they pass through a restriction. NE rupture is monitored by analysing the nuclear NLS-GFP intensity compared to the H2B signal.

### 5.3. Inducing Nuclear Envelope Ruptures

Non-transformed wild-type cell lines rarely show spontaneous NE ruptures [71,89]. However, NE ruptures can be induced by depleting lamins, expressing mutant lamin proteins, compromising the organisation of the peripheral heterochromatin or by viral infection. NE ruptures can also be triggered by mechanical stress. Techniques include constricted migration [6,7], laser ablation [83,86], cell compression between coverslips [76] and micro-indentation using Atomic Force Microscopy [75]. For more information on these and other methods to study mechanical properties of the nucleus see Hobson et al., 2021 [90]. It is also interesting to note that lipid-based transfection can increase membrane rupture frequency [80].

Inducing ruptures is particularly important when analysing specific events in the NE repair process, due to the fast nature of the repair process (a few minutes). High temporal resolution is therefore essential to catch specific events. It is important to note that it remains unclear how some of these methods accurately mimic and therefore reflect NE rupture repair in a physiological cellular context. Laser ablation for example could lead to additional effects such as DNA damage due to the high laser powers used. However, some papers do use multiple complementary methods showing that the rupture repair mechanisms are independent of the method used to induce rupture [7,83].

## 6. Concluding Remarks

The past decades have seen an increase in the quantity of research on the mechanobiology of the nucleus, due to its emerging links with diseases such as cancer and premature aging syndromes. Until recently, most analyses of nuclear abnormalities relied on manual scoring, which is low throughput and subjected to observer bias. With the recent advances in automated microscopy and high-throughput screening approaches, there has been a clear need for automated quantification of nuclear shape and NE defects. Here, we have discussed different nuclear abnormalities and published analysis pipelines used to quantify these. We have also summarised several metrics used for the quantification of nuclear shape. Furthermore, we have presented some new automated image analysis pipelines that we used for the analysis of large datasets. Undoubtedly, the development of open-source image analysis software such as CellProfiler will lead to the discovery of other diseases and processes involving nuclear biology, and will therefore contribute to further understanding of complex nuclear processes.

## Figures and Tables

**Figure 1 cells-11-00347-f001:**
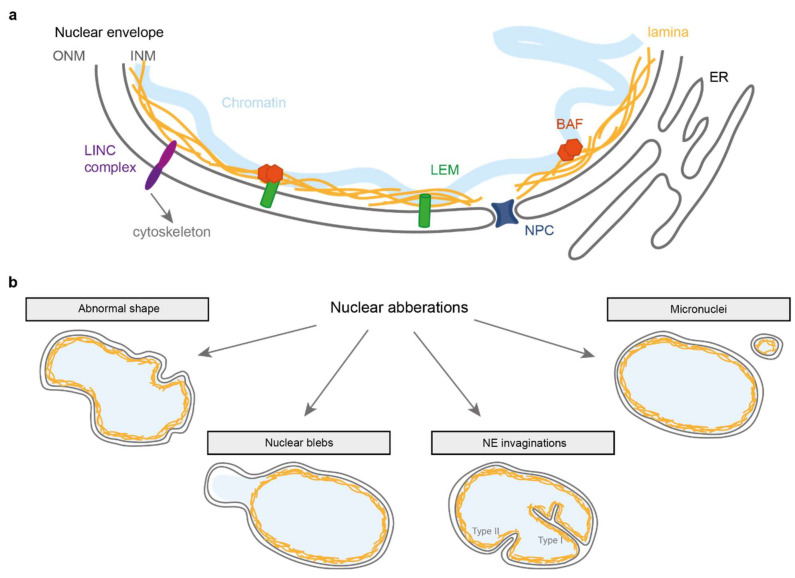
Simplified representation of the nuclear envelope and associated abnormalities. (**a**) The nuclear envelope consists of two membranes: the inner and outer nuclear membrane (INM and ONM respectively). Underlying the INM is the lamina composed of lamin A, B and C protein filaments that provide structural support to the nucleus. At the INM are members of the LEM domain protein family that can bind to the lamina directly or via BAF, a protein that binds DNA and the lamina. Spanning the NE are LINC complexes that connect the nuclear interior to various cytoskeletal components. (**b**) Schematic overview of nuclear aberrations discussed in this review.

**Figure 2 cells-11-00347-f002:**
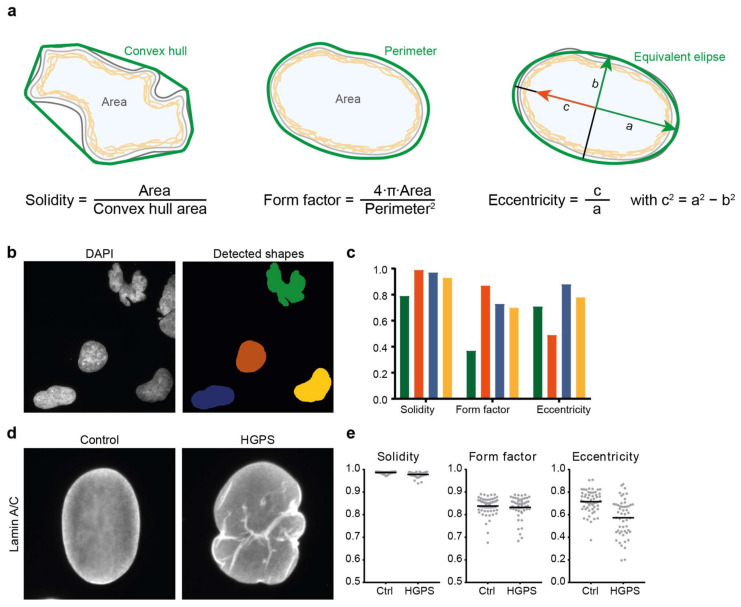
Commonly used parameters to describe nuclear shape. (**a**) Visual representation and mathematical definition of 3 routinely used nuclear shape descriptors: solidity, form factor (or circularity) and eccentricity. Values for each of these range from 0 to 1. (**b**,**c**) Example of nuclear shape analysis on DAPI-stained nuclei in fibroblasts derived from a patient with Nestor-Guillermo progeria syndrome (NGPS). Nuclear shapes in (**b**) were detected in CellProfiler; (**c**) Comparison of the values of three nuclear shape descriptors for the four detected nuclei shown in (**b**), with the colours of the bars corresponding to the colours of the nuclear masks in (**b**). Elongated shapes are better distinguished from rounder shapes by eccentricity than by solidity, while the highly lobulated nuclear shape is well distinguished using solidity and form factor. (**d**) Lamin A/C staining of a representative nucleus in a healthy donor fibroblast and a nucleus in a fibroblast from a patient with HGPS. The HGPS nucleus shows lamin invaginations, absent in the control nucleus, and a bulged nuclear contour. (**e**) Analysis of >50 nuclei in control and HGPS fibroblasts of similar cell passage yields significant differences for the nuclear solidity (*p* < 0.001) and eccentricity (*p* = 0.0003) but not for the form factor (*p* = 0.24). More elaborate nuclear shape analyses such as mean negative curvature calculation or elliptic Fourier transform (see text for details) are also able to distinguish control from HGPS nuclei.

**Figure 3 cells-11-00347-f003:**
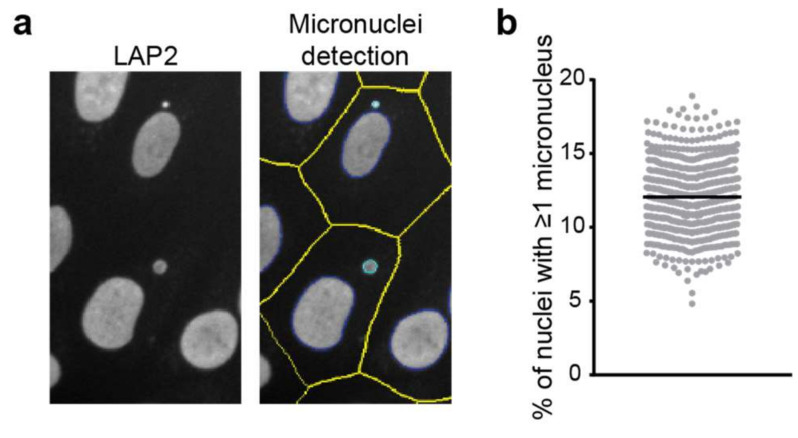
Detection of micronuclei in LAP2-stained images acquired on a high-content microscope. (**a**) Example of micronuclei detection in automated microscopy. Defined nuclear contours are indicated in dark blue, micronuclei in cyan and cell boundaries in yellow. See details of the analysis in the main text. (**b**) Quantitation of micronuclei frequency on images acquired on a high-content microscope in hTERT immortalised control fibroblasts in multiple biological replicates.

**Figure 4 cells-11-00347-f004:**
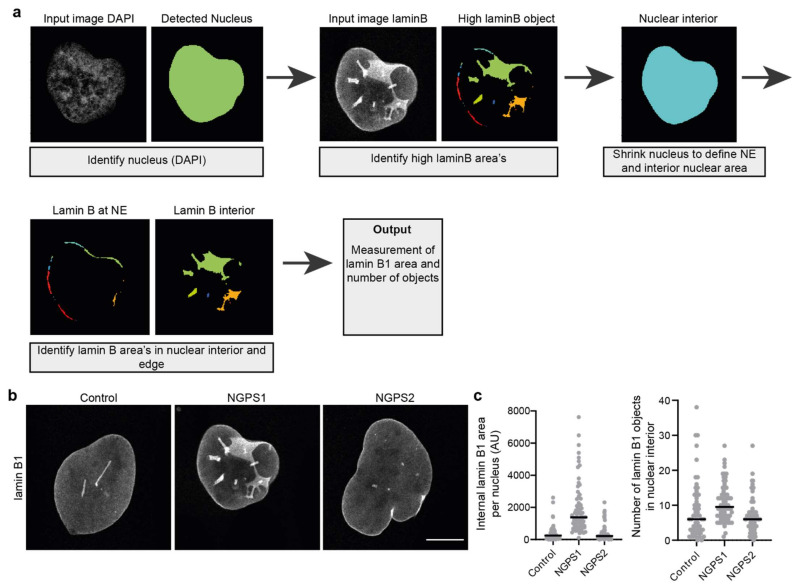
Quantification of nuclear envelope invaginations. (**a**) Simplified pipeline and example showing detection of nuclear invaginations in CellProfiler. (**b**) Representative immunofluorescence images of control and two NGPS patient-derived fibroblast cell lines (NGPS1 = NGPS5796 and NGPS2 = NGPS5787, gift from C. Lopez-Otin) stained for lamin B1. Scale bar 10µm. (**c**) Examples of intranuclear lamin B1 area quantification and of the number of detected lamin B1 objects in the nuclear interior of cells represented in (**b**) using the described analysis pipeline.

**Figure 5 cells-11-00347-f005:**
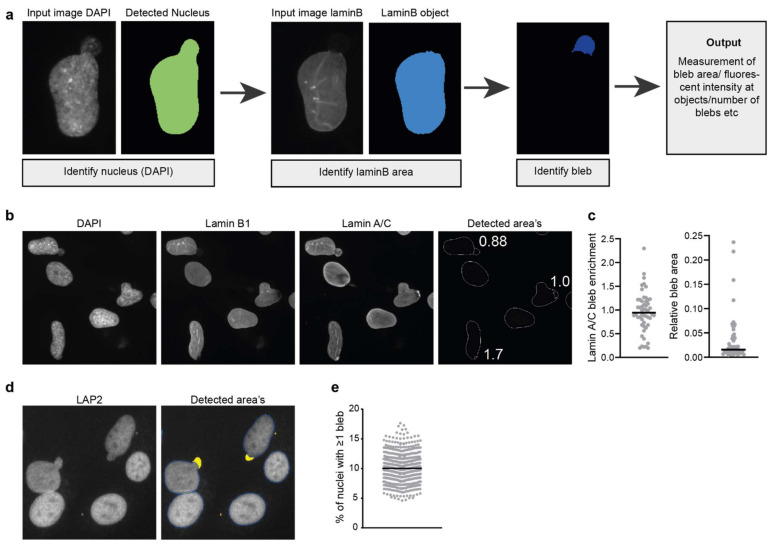
Detection and analysis of nuclear blebs. (**a**) Simplified pipeline and example showing the detection of nuclear blebs in CellProfiler. (**b**) Example image showing the endogenous staining of lamin B1 and lamin A/C in human fibroblast cells and the detected area’s using the pipeline described in (**a**). Numbers indicate the lamin A/C enrichment calculated for the detected blebs (laminA/C^bleb^/laminA/C^laminBobject^) (**c**) Example showing the quantification of lamin A/C enrichment at blebs in human fibroblast and the relative bleb area (area^bleb^/area^nucleus^) using the described pipeline. (**d**) Nuclear bleb detection in LAP2-stained nuclei in hTERT immortalised control fibroblasts in images acquired on a high-content microscope. The detection algorithm is based on the micronuclei detection algorithm (see text and Figure 3); however, the algorithm now selects nuclear blebs (yellow overlay) for analysis and rejects micronuclei (orange overlay) based on their size and distance to the primary nucleus (dark blue contour). (**e**) Quantification of nuclear bleb occurrence frequency in hTERT immortalised control fibroblasts in multiple biological replicates using automated nuclear bleb detection shown in (**d**).

## Data Availability

The presented Cell Profiler pipelines will be uploaded on the Cell Profiler website and be made freely available to the readers upon publication.

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
