# Peer review of "Current Methods and Pipelines for Image-Based Quantitation of Nuclear Shape and Nuclear Envelope Abnormalities"

_cells, 2022, doi:10.3390/cells11030347_

Round 1

Reviewer 1 Report

In my opinion, this manuscript is an excellent review paper. Particularly stressing that changes in the morphology of the cell nucleus may play a new and significant role in diagnostics. It turns out that such changes may be indicative of cancer or aging syndrome, especially today when, for example, recent advances in automated microscopy can be successfully used for similar purposes. 

Author Response

We would like to thank this reviewer for his/her very positive feedback on our manuscript. We are glad that he/she finds it a useful resource to the field.

Reviewer 2 Report

This review by Janssen et al., provides an introduction to the nuclear structures that help mediate nuclear shape, describes how nuclear shape is impacted by various pathologies, comprehensively focuses on the methods used to study and quantify nuclear shape, and finishes with an overview of studying blebs and ruptures. Overall, it is a well assembled and informative review. The authors could consider including direct links to software that are mentioned and/or use software names to facilitate those seeking to employ such methods, although the provided references are likely sufficient.

Minor comments:

In the introduction, perhaps mention reported roles for the chromatin itself in creating mechanical stability for the nucleus.

Line 31: Are there really lamin-like proteins in all non-metazoans, even in yeast?

In section 2.2 on nuclear shape it seems like one parameter that is not discussed but is perhaps most critical to nuclear area, diameter and perimeter, all 2D measurements as indicated, would be nuclear rounding. The size, i.e. volume, of the nucleus can remain the same but the surface area be increased, e.g. when flattened, giving a false sense of nuclear size changes in 2D. Many factors can influence this including substrate rigidity, confluency, differentiation state, and time since plating.

Micronuclei adjacent to the nucleus can occur and be difficult to distinguish from blebs, especially if reliant only on DNA labeling.

It might be worth mentioning sufficient exposure times with DNA stains can be important to detecting blebs where there is less DNA than in the rest of the nucleus. Overexposing can be used to account for these situations.

Author Response

We would like to thank the reviewer for taking the time to read our review and for the positive feedback and useful comments. With regard to the minor comments suggested by this reviewer, we have carefully taken them into consideration, making adjustments to the text as suggested.

In particular:

a/ We have added in the introduction (line 35) that chromatin itself can provide mechanical stability to the nucleus, and have added a reference to this.

b/ On line 31 we have removed “lamin-like proteins in non-metazoans” as indeed from current research it is unclear whether or not yeast has a lamin-like network underlying the NE. It has been shown that plants do indeed have lamin-like proteins that form a network at the nucleoplasmic side of the NE, but we feel that going into this level of detail is beyond the scope of our review.

c/ In section 2.2. we have included a sentence on nuclear rounding, how the same 3D nuclear volume can correspond to different 2D nuclear area, and how nuclear size can also be influenced by extra-cellular factors.

d/ We agree with the reviewer that micronuclei adjacent to the nucleus can occur. Therefore, we have now added a sentence on line 407 to specify that these micronuclei objects can be removed from the detected blebs by filtering for size or shape.

e/ We have included a sentence indicating that blebs often contain less DNA than the rest of the nucleus, and that therefore, care should be taken in setting the DNA detection threshold. We are shying away from suggesting over-exposing as that is generally not compatible with quantitative imaging; while it would be OK to “see” DNA in the blebs it would likely distort the nuclear contour and nucleus-associated measurements.